# Does the pressure to fill journal quotas bias evaluation?: Evidence from publication delays and rejection rates

**Brian Park**[1]*, **Eunhee Sohn**[2], **Soohun Kim**[3]

**1** Department of Managerial Sciences, J. Mack Robinson College of Business, Georgia State University, Atlanta, Georgia, United States of America, **2** Strategy and Innovation Area, Scheller College of Business, Georgia Institute of Technology, Atlanta, Georgia, United States of America, **3** Finance Area, KAIST College of Business, Korea Advanced Institute of Science and Technology, Seoul, Republic of Korea

* bpark@gsu.edu

**Data Availability Statement:** All the APA journal data are available from the website; https://www.apa.org/pubs/journals/statistics.

**Funding:** The author(s) received no specific funding for this work.

## Abstract

Although the peer review system of academic journals is seen as fundamental to scientific achievement, a major threat to the validity of the system is a potential evaluation bias resulting from constraints at the journal level. In this study, we examine how the time pressure to maintain a fixed periodical quota for journal publication can influence a journal editor's decision to accept or reject a paper at any given point in time. We find that an increase in publication backlog, proxied as the average delay between paper acceptance and print publication, is correlated with an increase in the subsequent rejection rates of new submissions. Our findings suggest that time pressures inherent in the peer review system may be a source of potential evaluator bias, calling for a need to reconsider the current quota system.

## Introduction

Modern scientific research blossoms in the form of publications. Today's peer review process serves as the backbone of modern science, resting on the postulate that publication validates the quality of scientific research [1–3]. Indeed, the idealized view of academic researchers is that submitted manuscripts are assessed by impartial evaluators based purely on manuscript quality, which is proportional to the likelihood of publication [3–5]. Anecdotal evidence suggests, however, that the academic journal review process persistently reflects errors in its acceptance of mediocre papers [6, 7] and its rejection of seminal research for publication [8, 9]. Furthermore, these systematic errors may be attributed at least partially to administrative constraints that affect the selection process [10–12].

Given that most academic journals take the form of printed periodicals with fixed paper quotas published at regular time intervals (e.g., weekly, monthly, quarterly, yearly), editors are expected to accept a consistent number of papers for publication at each interval. Editors, who have been called the "gatekeepers of science" [1, 13], wish to obtain high-quality papers and evaluate submitted manuscripts based purely on quality. However, their manuscript evaluation is also subject to the pressure to find a sufficient number of papers in time for the next regular

**Competing interests:** The authors have declared that no competing interests exist.

issue. Thus, if an editor has a predetermined paper quota and an insufficient number of accepted papers for the next issue, the pressure to find new papers to fill that quota may override the objective of maximizing the quality of the accepted papers.

In this study, we examine how such time pressures may have an unintended impact on the editor's acceptance decisions for submissions. We empirically proxy the number of accepted but not yet published papers using the publication delay between paper acceptance and publication in print. Our analysis provides evidence that when the number of accepted but not yet published papers is sufficient to meet the quota for the next regular issues of the print journal, editors are more likely to reject submissions because the pressure to meet the quota within a given time is low.

Time pressure is a ubiquitous phenomenon in our world. Our daily decision making in contexts such as stock trading, shopping, and job searching occurs within limited time constraints that often press us to make quick decisions [14]. A number of researchers in the social sciences provide ample evidence that time pressure plays a significant role in decision making processes and risk-taking behavior. The existing literature documents how time pressure leads to increased risk aversion to losses [15, 16] and reductions in the impact of product recommendations on consumer choices [17]. Some economists studying the effect of time pressure on successful bargaining have found that a high percentage of agreements are reached very close to the deadline [18]. Others have demonstrated the opposite effect in an ultimatum game, such that subjects are more likely to reject an offer under time pressure [19].

Like those in many contexts, then, editors of academic journals are constantly exposed to time pressure as they must meet their quota for quality papers at regular intervals. If editors fail in this task, they may be viewed by the editorial board, associate editors, and authors as performing poorly, and in some cases, be subject to penalties. Such pressure is likely to be augmented when editors find themselves short of accepted papers for the next regular issues of their journal, a situation in which they may consider accepting papers of less quality that might otherwise be rejected. When editors are subject to less time pressure, however, they may increase their quality threshold for incoming papers and eventually reject even high-quality papers that would have been accepted had they been submitted at a different time.

## Model

Based on recent literature on time pressure and task quality, we developed a mathematical model to specify how an editor's quality threshold for paper acceptance varies according to time pressure, a proxy for which is the backlog of accepted, yet unpublished, papers. The model assumes that an editor seeks to maximize the aggregate quality of papers in a journal according to the constraints of a fixed paper quota while papers of random quality are submitted to the journal at a constant probability. In fact, under such constraints, the greater the number of papers accepted, yet waiting to be published, the lower the possibility for a subsequently submitted paper of equal quality to be accepted. In the formal descriptions of the model that follow, the editor accepts, at most, $K$ papers for a journal that needs to be published at calendar time $T$. The goal of the editor is to maximize the aggregate quality of papers in the journal. At $t \in [0, T]$, the editor receives a paper with the arrival rate of $\lambda$. The distribution of paper quality is known to follow $F(q)$, which does not depend on $t$. If a paper with the quality $q$ arrives at time $t$, the editor accepts or rejects the paper based on the given quality of the paper. Let $k$ denote the number of papers which are already accepted. The editor needs to select $(k + 1)$-th paper for the journal. We are interested in the effect of $k$, the number of prepublication accepted papers, on the editor's time pressure and on the probability of rejection of a received paper, which is a candidate for the $(k + 1)$-th paper.

**Proposition 1**. *When $k' > k$, $\underline{q}(k', t) \geq \underline{q}(k, t)$.*

It is worth highlighting that this proposition directly confirms our main argument because $\underline{q}(k + 1, t) \geq \underline{q}(k, t)$ implies that the probability of acceptance for the *(k + 1)*-th paper is lower than that of *k*-th paper because $\Pr(q > \underline{q}(k + 1, t)) \leq \Pr(q > \underline{q}(k, t))$, which implies that the editor's quality threshold for manuscript acceptance changes according to time pressure, which is measured by the number of accepted papers. The details for the proof of the proposition is included in our S1 Appendix.

## Materials and methods

We empirically tested our argument using the 2004–2017 annual publication data for 54 journals affiliated with the American Psychological Association (APA) (Table 1). All these journals disclose complete information on the rejection rates, annual number of received/accepted/pending papers, and number of items published in print (summary statistics and histograms are displayed in Table 2 and Fig 1, respectively). While the data does not reveal the average number of publication backlogs *per se* in a given year, the data instead provides the average publication delay of a given year in months, that is, the average difference in months between the dates of final paper acceptance and publication in print. We used this delay measure as a proxy for the backlog of accepted, but not formally published, papers (these are often referred to as forthcoming or in-press papers and do not have volume and issue information assigned).

**Table 1. Journal list.**

| | |
|---|---|
| American Journal of Community Psychology | Journal of Pediatric Psychology |
| American Psychologist | Journal of Personality and Social Psychology |
| Analyses of Social Issues and Public Policy | Law and Human Behavior |
| Behavioral Neuroscience | Military Psychology |
| Clinical Psychology: Science and Practice | Neuropsychology |
| Cultural Diversity and Ethnic Minority Psychology | Professional Psychology: Research and Practice |
| Developmental Psychology | Psychoanalytic Psychology |
| Educational Psychologist | Psychological Assessment |
| Emotion | Psychological Bulletin |
| Experimental and Clinical Psychopharmacology | Psychological Methods |
| Group Dynamics: Theory, Research, and Practice | Psychological Review |
| Health Psychology | Psychological Services |
| History of Psychology | Psychological Trauma |
| JEP: Animal Learning and Cognition | Psychology and Aging |
| JEP: Applied | Psychology of Addictive Behaviors |
| JEP: General | Psychology of Aesthetics, Creativity and the Arts |
| JEP: Human Perception and Performance | Psychology of Men and Masculinity |
| JEP: Learning, Memory, and Cognition | Psychology of Religion and Spirituality |
| Journal of Abnormal Psychology | Psychology of Women Quarterly |
| Journal of Applied Psychology | Psychology, Public Policy, and Law |
| Journal of Clinical Child and Adolescent Psychology | Psychotherapy |
| Journal of Comparative Psychology | Rehabilitation Psychology |
| Journal of Consulting and Clinical Psychology | Review of General Psychology |
| Journal of Consumer Psychology | School Psychology Quarterly |
| Journal of Counseling Psychology | Sport, Exercise, and Performance Psychology |
| Journal of Educational Psychology | Teaching of Psychology |
| Journal of Family Psychology | The Counseling Psychologist |

**Table 2. Summary statistics.**

|  | Mean | s.d. | Min | Max |
|---|---:|---:|---:|---:|
| Rejection rates | 0.717 | 0.132 | 0.310 | 1.000 |
| Publication delay (log) | 1.708 | 0.429 | 0.405 | 2.773 |
| Number of papers accepted | 70.012 | 41.136 | 4 | 240 |
| Number of papers received | 311.462 | 230.842 | 47 | 1101 |
| Number of papers pending | 75.917 | 69.169 | 0 | 482 |
| Count of items (log) | 4.137 | 0.588 | 1.946 | 5.533 |
| Journal impact factor (log) | 0.781 | 0.702 | -1.737 | 2.808 |

Then, we empirically examined the relationship between publication delays and rejection rates.

In most disciplines only a very small number of manuscripts is accepted for publication on the first submission, and most editors solicit at least one revision from authors. To capture the average review time for APA journals, we randomly selected five papers per publication year from each of APA-affiliated journals, all of which disclosed the specific dates of the first submission and subsequent acceptance for each paper from 2004 through 2017. The average review time was 285 days and the standard deviation was 195.41 days, ranging from a few months to more than two years. Given that the review process period of most of the submitted manuscripts was shorter than two years (less than 730 days), we averaged the publication delays at one and two years prior to the focal year, took the logarithm, and used it as our explanatory variable to see its effect on rejection rate in year $t$.

The reasoning behind this operationalization is that journals usually report their rejection rate in year $t$ as 1 minus acceptance rate, which is the number of accepted papers in year $t$ divided by the number of received papers in year $t$. This means that most of the rejections that

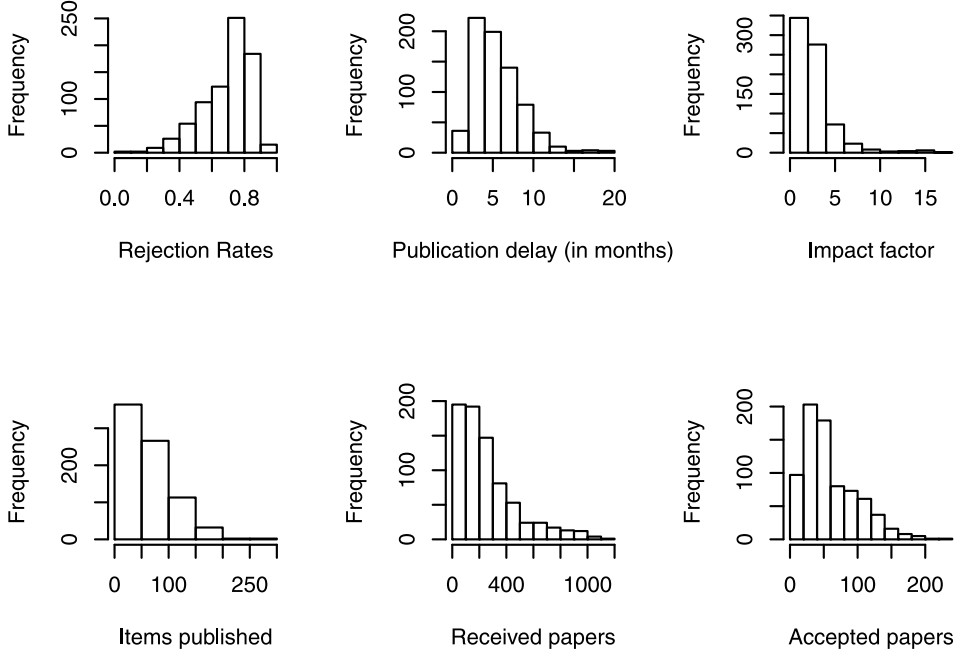

**Fig 1. Histograms of the key variables.**

occur during the review process, which takes up to two years on average, do *not* immediately factor into the acceptance/rejection rate of the same year. Let us assume that during an evaluation process, an editor rejects a paper submitted in year *t-1* or year *t-2* based on the publication delay of that year. This decision would, in effect, affect the rejection rate of year *t*, given that the counterfactual final acceptance would have occurred in year *t*. Although the 54 APA-affiliated journals in question reported their rejection rates, each academic journal used a different method for calculating the rate. Therefore, for a robustness check, we also tested the extent to which publication delays affect the absolute number of accepted papers. We included two key variables as controls in our analysis. One is the count of items published in print (log) because it indicates the capacity for printed papers of each journal, and the other is the journal impact factor without self citations (log) drawn from Thomson Reuters, since high-quality journal editors may be less concerned about publication delays. Lastly, we included in the model fixed effect dummy variables for journal and year.

## Results

Prior to the main analysis, we plotted the relationship between rejection rate and publication delay (Fig 2). The least-square fitted line in this figure shows that rejection rate is positively associated with publication delay, though a caveat is needed, as this estimation fails to consider the control variables and the fixed effects. Showing the least-square line, however, helps us capture the relationship between the two variables in an illustrative manner. In our main analysis, ordinary least squares (OLS) regression with the control variables, as well as year/journal fixed effects, indicates a positive relationship between publication delay and subsequent rejection rate (see Fig 3; a plot of the coefficients of Model 2 in Table 3) regardless of the control variables (Models 1 and 2 in Table 3).

The log–log regression coefficient estimates showed that a 1% increase in a journal's publication delay is correlated with about a 0.1% increase in the journal's rejection rate, holding

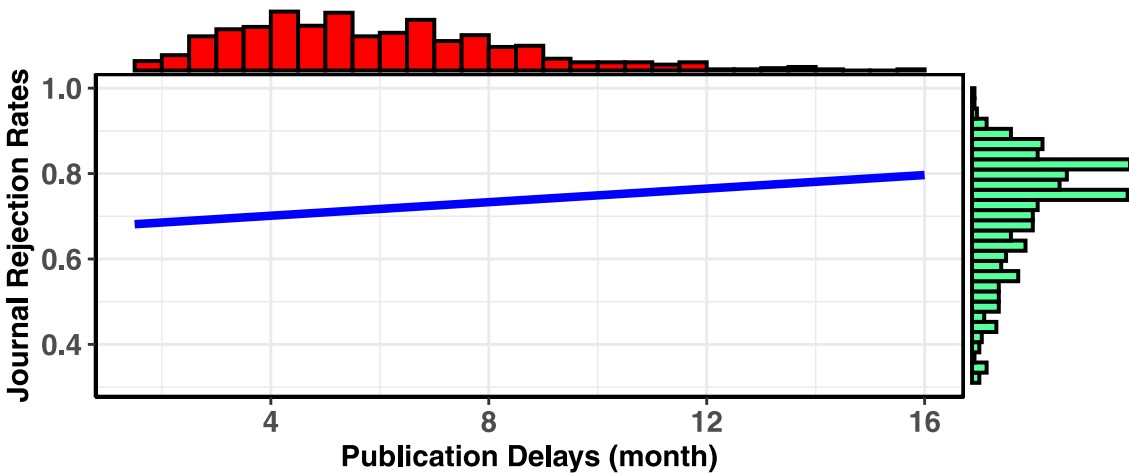

**Dataset: APA journal 2004–2017**

$t(509) = 3.44$, $p = 0.001$, $r_{\text{Pearson}} = 0.15$, $\text{CI}_{95\%}$ [0.06, 0.23], $n_{\text{pairs}} = 511$

In favor of null: $\log_e(\text{BF}_{01}) = -3.12$, $r_{\text{Cauchy}}^{\text{JZS}} = 0.71$

**Fig 2. Visualization of publication delays and rejection rates in APA-affiliated journals.**

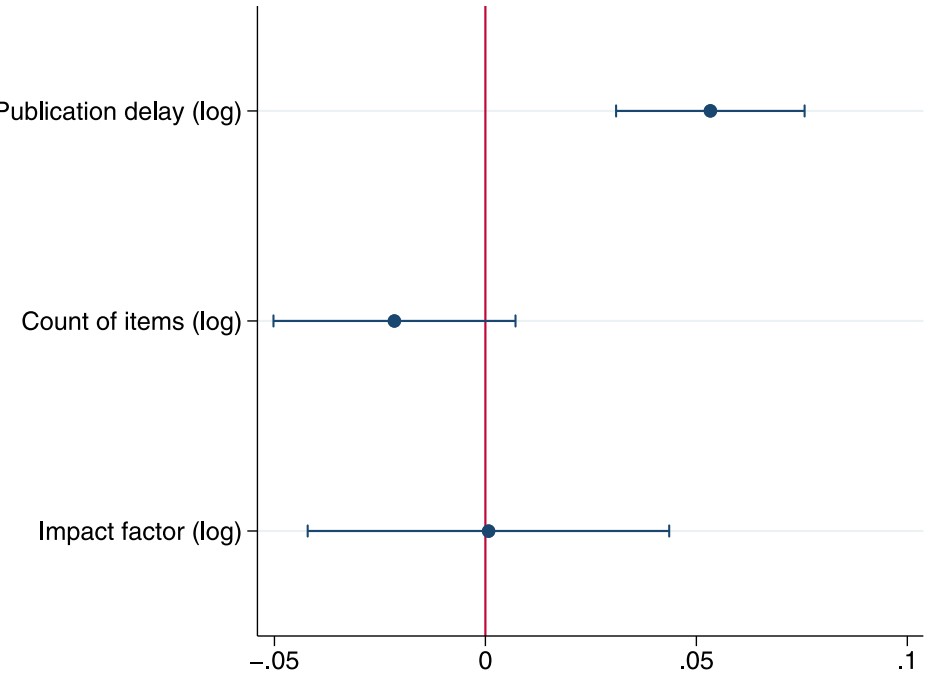

**Fig 3. A plot of the coefficients in the main model (Model 2 in Table 3).**

**Table 3. The effect of publication delay on APA rejection rates.**

|  | Model 1 | Model 2 | Model 3 | Model 4 | Model 5 | Model 6 |
|---|---|---|---|---|---|---|
| Outcome variable | Rejection rate (log) | | Rejection rate | | Rejection rate | |
| Estimation | OLS | | OLS | | Tobit | |
| Publication delay (log) Count of items (log) Impact factor (log) | 0.058** | 0.084*** | 0.038** | 0.053*** | 0.038*** | 0.053*** |
|  | (0.018) | (0.018) | (0.011) | (0.011) | (0.011) | (0.011) |
|  |  | -0.030 |  | -0.022 |  | -0.022 |
|  |  | (0.021) |  | (0.014) |  | (0.014) |
|  |  | -0.004 |  | 0.001 |  | 0.001 |
| Journal fixed effect$_j$ Year fixed effect$_t$ Constant |  | (0.035) |  | (0.021) |  | (0.021) |
|  | Yes | Yes | Yes | Yes | Yes | Yes |
|  | Yes | Yes | Yes | Yes | Yes | Yes |
|  | -0.470*** | -0.394*** | 0.643*** | 0.702*** | 0.673*** | 0.727*** |
|  | (0.042) | (0.091) | (0.025) | (0.059) | (0.027) | (0.062) |
| R-squared | 0.084 | 0.106 | 0.096 | 0.119 |  |  |
| Count of Journals | 60 | 54 | 60 | 54 | 60 | 54 |
| N | 559 | 511 | 559 | 511 | 559 | 511 |

Coefficients are reported with robust standard errors clustered by each journal in parentheses.

All models include journal and year fixed effects.

$^+p<0.10$,

$^*p<0.05$,

$^{**}p<0.01$,

$^{***}p<0.001$.

**Table 4. The effect of publication delay on the number of received/pending/accepted papers (OLS).**

| | Model 1 | Model 2 | Model 3 |
|---|---|---|---|
| **Outcome variable** | **Number of accepted** | **Number of received** | **Number of pending** |
| | **papers (log)** | **papers (log)** | **papers (log)** |
| Publication delay (log) | -0.113** | 0.002 | -0.161 |
| | (0.038) | (0.040) | (0.097) |
| Count of items (log) | 0.537*** | 0.441*** | 0.269* |
| | (0.092) | (0.081) | (0.121) |
| Impact factor (log) | 0.035 | -0.000 | 0.023 |
| | (0.085) | (0.051) | (0.118) |
| Journal fixed effect$_j$ | Yes | Yes | Yes |
| Year fixed effect$_t$ | Yes | Yes | Yes |
| Constant | 1.889*** | 3.413*** | 3.052*** |
| | (0.354) | (0.298) | (0.452) |
| R-squared | 0.265 | 0.531 | 0.098 |
| Count of Journals | 54 | 54 | 54 |
| N | 511 | 513 | 510 |

Coefficients are reported with robust standard errors clustered by each journal in parentheses.

All models include journal and year fixed effects.

$^+p<0.10$,

$^*p<0.05$,

$^{**}p<0.01$,

$^{***}p<0.001$.

other variables constant. We also used the absolute number of accepted papers (log) as our outcome variable. The finding holds with the log of the number of accepted papers as our outcome variable (Model 1 in Table 4); publication delay is negatively associated with the subsequent number of accepted papers. Critics may argue that authors can strategically change their submission time by considering their target journal's publication delay. If this argument is true, we should find a relationship between the number of received papers and publication delay. We did not, however, find any significant impact of publication delay on the (log) number of received papers (Model 2 in Table 4) or the (log) number of pending papers (Model 3 in Table 4). This shows that the positive correlation between publication delay and rejection rate is not driven by the increasing popularity of some journals. We also estimated our models with the absolute number of accepted / received / pending papers as our outcome variable (Table 5) in the negative binomial estimation, with consistent results.

We further explored our results by splitting the publication delay variable by year, that is, a one-year publication delay and two-year publication delay prior to the focal year. We consistently found that both publication delay variables have a significant impact on rejection rate and the number of accepted papers (Table 6). In each model we compared the coefficient of one-year and two-year publication delays prior to the focal year and did not find a significant difference. Finally, we tested whether publication delay is associated with the subsequent quality of the journal. If an editor is less likely to accept low-quality papers when enough papers have been accepted to meet the regular quota, we should see a positive relationship between publication delay and journal quality, as we predicted in our mathematical model. We regressed the journal impact factor on publication delay in the ordinary least squares (Model 1 in Table 7) or Tobit estimation because the journal impact factor is left truncated at 0 (Model 2

**Table 5. The effect of publication delay on the number of received/pending/accepted papers (Negative Binomial).**

| | Model 1 | Model 2 | Model 3 |
|---|---|---|---|
| Outcome variable | Number of accepted | Number of received | Number of pending |
| | papers | papers | papers |
| Publication delay (log) | -0.130*** | 0.003 | -0.160+ |
| | (0.032) | (0.035) | (0.086) |
| Count of items (log) | 0.572*** | 0.436*** | 0.283* |
| | (0.086) | (0.076) | (0.123) |
| Impact factor (log) | 0.029 | 0.003 | 0.018 |
| | (0.072) | (0.053) | (0.108) |
| Journal fixed effect$_j$ | Yes | Yes | Yes |
| Year fixed effect$_t$ | Yes | Yes | Yes |
| Constant | 1.853*** | 3.425*** | 1.824*** |
| | (0.354) | (0.302) | (0.470) |
| Log likelihood | -2077.812 | -2608.358 | -2257.808 |
| Count of Journals | 54 | 54 | 54 |
| N | 511 | 513 | 510 |

Coefficients are reported with robust standard errors clustered by each journal in parentheses.

All models include journal and year fixed effects.

+$p<0.10$,

*$p<0.05$,

**$p<0.01$,

***$p<0.001$.

in Table 7), and we found a significant positive relationship between the two. This finding suggests that a low publication delay period (i.e., a lack of accepted, yet unpublished, papers) may decrease the quality of the journal because the editor may assume a more lenient view of newly submitted manuscripts in order to fill the quota. Finally, we performed additional analysis that supports robustness of our results. We simulated monthly data to examine the magnitude of the bias from using aggregated data at the year level. Our simulated data show that on average, we obtain a coefficient value, similar to our empirical observation. Also the simulated data show that we find a substantial attenuation when measuring the quantitative relationship between year-based rejection rates and publication delay.

## Discussion

This paper demonstrates that the academic journal review process is influenced by the time pressure to publish a fixed number of quality papers on a particular schedule, a phenomenon that arises from an academic journal's periodical nature and quota constraints. With an obligation to publish a consistent number of papers at regular intervals, an increase or decrease in the stock of accepted papers (proxied in this study as publication delay) may influence a journal editor's time pressure and view of incoming manuscripts. Our finding suggests that even the evaluation of research by an expert academic editor is not completely independent of the time pressure to meet the journal quota. This study has several implications. For authors, publication delays—a proxy for the backlog of accepted but not formally published papers—may work as a factor motivating authors' choice of journals, given that it influences rejection rates. For editors, we strongly recommend that academic journals publicly disclose their publication

**Table 6. The effect of publication delay on rejection rate / number of accepted papers.**

| | Model 1 | Model 2 |
|---|---|---|
| Outcome variable | APA rejection rates | Number of accepted papers |
| Estimation | OLS | Negative Binomial |
| Publication delay (log)$_{t-1}$ | 0.017+ | -0.068* |
| | (0.009) | (0.031) |
| Publication delay (log)$_{t-2}$ | 0.033*** | -0.055+ |
| | (0.009) | (0.030) |
| Count of items (log) | -0.020 | 0.573*** |
| | (0.014) | (0.085) |
| Impact factor (log) | 0.000 | 0.027 |
| | (0.021) | (0.071) |
| Journal fixed effect$_j$ | Yes | Yes |
| Year fixed effect$_t$ | Yes | Yes |
| Constant | 0.708*** | 1.824*** |
| | (0.058) | (0.352) |
| Count of Journals | 54 | 54 |
| N | 511 | 511 |

Coefficients are reported with robust standard errors clustered by each journal in parentheses.

All models include journal and year fixed effects.

+$p < 0.10$,

*$p < 0.05$,

**$p < 0.01$,

***$p < 0.001$.

**Table 7. The effect of publication delay on impact factor.**

| | Model 1 | Model 2 |
|---|---|---|
| Outcome variable | Impact factor | |
| Publication delay (log) | 0.231* | 0.231* |
| | (0.103) | (0.102) |
| Count of items (log) | 0.021 | 0.021 |
| | (0.157) | (0.155) |
| Journal fixed effect$_j$ | Yes | Yes |
| Year fixed effect$_t$ | Yes | Yes |
| Constant | 1.840** | 0.367 |
| | (0.627) | (0.648) |
| Count of journals | 54 | 54 |
| N | 516 | 516 |

Coefficients are reported with robust standard errors clustered by each journal in parentheses.

All models include journal and year fixed effects.

+$p < 0.10$,

*$p < 0.05$,

**$p < 0.01$,

***$p < 0.001$.

delays, or the number of pre-published accepted papers, to help potential authors better time their submissions.

A number of researchers have suspected that publication outcome can be affected by the backlog of papers, but there is only one empirical paper [20], to our knowledge, which has tested this backlog impact, and that in an indirect manner. Although academic journals have differing amounts of space available for accepted papers, the study mentioned argued that there is no evidence showing that journal rejection rates are influenced by their space shortages. We are skeptical about this conclusion because, as the authors acknowledged, the challenge of measuring space shortage caused them simply to use the number of papers submitted to a journal as a proxy for its space shortage, which is then problematically correlated with rejection rates. We believe that this is a very first paper to *empirically* test and show the effect of the backlog of accepted papers or publication delays on subsequent rejection rates.

More broadly, we believe that it is an opportune time to reconsider the utility of the current quota system of most academic journals. A significant number of journals are still bound by the quota system of print publications despite their concurrent publication of electronic versions of their paper-based journals. We suggest that such a quota-based journal system exposes editors to the pressure of having a constant number of papers ready to print for every regular issue, thus subjecting submitted papers to an evaluation based not only on quality but also on the variable demand for new papers. This problem has been noted by other researchers, who suggest that space shortages (a variable similar to publication delay in this paper) influence journal review processes, though without empirical evidence to support this proposition.

Print periodicals have long been touted as one of the most useful forms for sharing academic findings with both researchers and the public on a regular basis. Yet, the cyclical demand for print-based periodicals may lead editors to fall into the selection error of accepting mediocre papers when in need of more content or rejecting quality papers when a sufficient backlog of already accepted papers exists. We contend, therefore, that it is time for editors to have a more flexible approach to page budgeting; rather than limiting the number of issues for a given period they might skip a regular issue when short of high-quality papers or add an issue when a backlog of accepted papers is high.

## Supporting information

**S1 Appendix.**
(PDF)

## Author Contributions

**Conceptualization:** Brian Park, Soohun Kim.

**Investigation:** Brian Park, Eunhee Sohn.

**Resources:** Brian Park.

**Writing – original draft:** Brian Park.

**Writing – review & editing:** Brian Park, Eunhee Sohn, Soohun Kim.

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
