## [Decision Letter · Decision Letter 0]

5 Dec 2019

PONE-D-19-25612

Does the Pressure to Fill Journal Quotas Bias Evaluation?: Evidence from Publication Delays and Rejection Rates

PLOS ONE

Dear Mr. Park,

Thank you for submitting your manuscript to PLOS ONE. After careful consideration, we feel that it has merit but does not fully meet PLOS ONE’s publication criteria as it currently stands. Therefore, we invite you to submit a revised version of the manuscript that addresses the points raised during the review process.

Two acknowledged experts in the field have commented on your paper. Overall, they find a certain potential in your work to be published, but several key concerns (most of them major), that include biasing sources, interpretations and statistical effects, need your revision and response, as you will see below in the feedback provided by our academic reviewers.

We would appreciate receiving your revised manuscript by Jan 19 2020 11:59PM. To enhance the reproducibility of your results, we recommend that if applicable you deposit your laboratory protocols in protocols.io, where a protocol can be assigned its own identifier (DOI) such that it can be cited independently in the future. For instructions see: http://journals.plos.org/plosone/s/submission-guidelines#loc-laboratory-protocols

We look forward to receiving your revised manuscript.

Kind regards,

Sergio A. Useche, Ph.D.

Academic Editor

PLOS ONE

Journal Requirements:

**When submitting your revision, we need you to address these additional requirements:**

**Please ensure that your manuscript meets PLOS ONE's style requirements, including those for file naming. The PLOS ONE style templates can be found at http://www.plosone.org/attachments/PLOSOne_formatting_sample_main_body.pdf and http://www.plosone.org/attachments/PLOSOne_formatting_sample_title_authors_affiliations.pdf**

 2. Please include a separate caption for each figure in your manuscript.

Reviewers' comments:

Reviewer's Responses to Questions

**Comments to the Author**

1. Is the manuscript technically sound, and do the data support the conclusions?

Reviewer #1: Partly

Reviewer #2: Partly

2. Has the statistical analysis been performed appropriately and rigorously? 

Reviewer #1: I Don't Know

Reviewer #2: Yes

3. Have the authors made all data underlying the findings in their manuscript fully available?

Reviewer #1: Yes

Reviewer #2: No

4. Is the manuscript presented in an intelligible fashion and written in standard English?

Reviewer #1: Yes

Reviewer #2: Yes

5. Review Comments to the Author

Reviewer #1: This was an interesting paper and it seemed correct. However, there is one important thing that the editor should check, and I would be interested to see if other reviewers had this concern as well. Note: I didn’t have time to review the mathematical model, and am mainly provided comments on the empirical data analysis.

As I understand it, the data are very coarse, and are measured on an annual basis. It would be much better to have data on a monthly basis, but I understand that might not be available. What would want to try to recover is the editor’s decision at a specific point in time t (i.e., what is the backlog at this time when he/she is evaluating a group of papers?). Since editors don’t make decisions at one point in time in the year, averaging across the year might generate some bias. I think the bias is probably going to attenuate the effect, but the authors need to more copiously and cogently deal with this issue of measurement error. This is particularly because the effect sizes seem very small to me (keep in mind these are log-log models, so 0.8% does not mean 0.8 percentage points). It could reflect going from 10% acceptance rate to 9.9%.

Further, some of the author decisions seemed sloppy, such as only using one randomly selected paper per year to estimate review time and publication delay. Why could they have not done this with more papers to get a better estimate? Poor measurement of this variable would also probably bias the estimates downward.

Reviewer #2: Does the Pressure to Fill Journal Quotas Bias Evaluation? Evidence from Publication Delays and Rejection Rates

This paper sets out to examine how time pressure to maintain a fixed periodical quota for journal publications can influence a journal editor’s decision to accept or reject a paper. The authors find that a 10% increase in publication backlog, is correlated with a 0.8% increase in the subsequent rejection rates of new submissions. This seems very intuitive: when an editor has more papers to choose from, (s)he will reject a higher percentage of submissions. This finding seems to be very much what one would expect. I doubt whether this is a very interesting finding to publish.

Subsequently, the authors suggest (p. 9) ‘that a low publication delay period, may decrease the quality of the journal because the editor may assume a more lenient view of newly submitted manuscripts in order to fill the quota.’ Results in Table 7 indeed show that publication delays have a significant positive effect on impact factor. However, more information should be given about the impact factor in general and how it is measured in this paper to determine whether the authors indeed prove what they claim to prove. What (in general) is the relation between time after publication and number of citations? Is this a normal distribution? If so, papers will have a relatively small effect on the impact factor of a journal in the first year after their publication. After how many years/what period will the effect of the quality of papers be felt in the impact factor of a journal? Also, how many years/months before an article was published, was it submitted? It might take quite some time before the quality of an article submitted in year t, published in year t+a, affects the impact factor of the journal in year t+a+b.

For this study: from which years were the impact factors taken? How many years difference is there between the year of impact factor and years used to calculate publication delay?

The relevance of the study mentioned on page 4, i.e. that a task completed near a deadline is of lower quality than one completed well before the deadline, to the study in this paper is not clear and needs to be explained better. The study mentioned on page 4 is about the effect of time pressure on the work someone is performing him/herself. The paper under review is about the effect of time pressure felt by the editor on the quality of accepted work done by others than the editor.

Figures 2 and 3: it is not clear to me how to read these figures. This has to be explained better.

6. PLOS authors have the option to publish the peer review history of their article (what does this mean?). If published, this will include your full peer review and any attached files.

Reviewer #1: No

Reviewer #2: No

[NOTE: If reviewer comments were submitted as an attachment file, they will be attached to this

---

## [Author Response · Author response to Decision Letter 0]

3 Jul 2020

All our responses are included in one of our documents "Responses to Reviewers"

---

## [Decision Letter · Decision Letter 1]

17 Jul 2020

Does the Pressure to Fill Journal Quotas Bias Evaluation?: Evidence from Publication Delays and Rejection Rates

PONE-D-19-25612R1

Dear Dr. Park,

We’re pleased to inform you that your manuscript has been judged scientifically suitable for publication and will be formally accepted for publication once it meets all outstanding technical requirements.

Kind regards,

Sergio A. Useche, Ph.D.

Academic Editor

PLOS ONE

Additional Editor Comments (optional):

Reviewers' comments:

Reviewer's Responses to Questions

**Comments to the Author**

1. If the authors have adequately addressed your comments raised in a previous round of review and you feel that this manuscript is now acceptable for publication, you may indicate that here to bypass the “Comments to the Author” section, enter your conflict of interest statement in the “Confidential to Editor” section, and submit your "Accept" recommendation.

Reviewer #1: All comments have been addressed

2. Is the manuscript technically sound, and do the data support the conclusions?

Reviewer #1: Yes

3. Has the statistical analysis been performed appropriately and rigorously? 

Reviewer #1: Yes

4. Have the authors made all data underlying the findings in their manuscript fully available?

Reviewer #1: Yes

5. Is the manuscript presented in an intelligible fashion and written in standard English?

Reviewer #1: Yes

6. Review Comments to the Author

Reviewer #1: This was an impressive revision. The authors conducted several additional analyses and data collection efforts to address my concerns. I assumed the editor believed my initial comments could be handled.

7. PLOS authors have the option to publish the peer review history of their article (what does this mean?). If published, this will include your full peer review and any attached files.

Reviewer #1: No

---

## [Editor Report · Acceptance letter]

30 Jul 2020

PONE-D-19-25612R1 

Does the Pressure to Fill Journal Quotas Bias Evaluation?: Evidence from Publication Delays and Rejection Rates 

Dear Dr. Park:

I'm pleased to inform you that your manuscript has been deemed suitable for publication in PLOS ONE. Congratulations! Your manuscript is now with our production department. 

Kind regards, 

on behalf of

Dr. Sergio A. Useche 

Academic Editor

PLOS ONE